# SARS-CoV-2 seroprevalence in three Kenyan health and demographic surveillance sites, December 2020-May 2021

Anthony O. Etyang[1]*, Ifedayo Adetifa[1,2], Richard Omore[3], Thomas Misore[3], Abdhalah K. Ziraba[4], Maurine A. Ng'oda[4], Evelyn Gitau[4], John Gitonga[1], Daisy Mugo[1], Bernadette Kutima[1], Henry Karanja[1], Monica Toroitich[1], James Nyagwange[1], James Tuju[1], Perpetual Wanjiku[1], Rashid Aman[5], Patrick Amoth[5], Mercy Mwangangi[5], Kadondi Kasera[5], Wangari Ng'ang'a[6], Donald Akech[1], Antipa Sigilai[1], Boniface Karia[1], Angela Karani[1], Shirine Voller[1,2], Charles N. Agoti[1], Lynette I. Ochola-Oyier[1], Mark Otiende[1], Christian Bottomley[1], Amek Nyaguara[1], Sophie Uyoga[1], Katherine Gallagher[1], Eunice W. Kagucia[1], Dickens Onyango[7], Benjamin Tsofa[1], Joseph Mwangangi[1], Eric Maitha[8], Edwine Barasa[9,10], Philip Bejon[1,10], George M. Warimwe[1,10], J. Anthony G. Scott[1,2], Ambrose Agweyu[1]

1 KEMRI-Wellcome Trust Research Programme, Kilifi, Kenya, 2 London School of Hygiene and Tropical Medicine, London, United Kingdom, 3 Kenya Medical Research Institute Centre for Global Health Research, Kisumu, Kenya, 4 African Population and Health Research Center, Nairobi, Kenya, 5 Ministry of Health, Nairobi, Kenya, 6 Presidential Policy and Strategy Unit, The Presidency, Government of Kenya, Nairobi, Kenya, 7 Department of Health, Kisumu County, Kenya, 8 Department of Health, Kilifi County, Kenya, 9 Health Economics Research Unit, KEMRI-Wellcome Trust Research Programme, Nairobi, Kenya, 10 Centre for Tropical Medicine and Global Health, Nuffield Department of Medicine, Oxford University, Oxford, United Kingdom

☯ These authors contributed equally to this work.
* Aetyang@kemri-wellcome.org

## Abstract

### Background

Most of the studies that have informed the public health response to the COVID-19 pandemic in Kenya have relied on samples that are not representative of the general population. We conducted population-based serosurveys at three Health and Demographic Surveillance Systems (HDSSs) to determine the cumulative incidence of infection with SARS-CoV-2.

### Methods

We selected random age-stratified population-based samples at HDSSs in Kisumu, Nairobi and Kilifi, in Kenya. Blood samples were collected from participants between 01 Dec 2020 and 27 May 2021. No participant had received a COVID-19 vaccine. We tested for IgG antibodies to SARS-CoV-2 spike protein using ELISA. Locally-validated assay sensitivity and specificity were 93% (95% CI 88–96%) and 99% (95% CI 98–99.5%), respectively. We adjusted prevalence estimates using classical methods and Bayesian modelling to account for the sampling scheme and assay performance.

**Data Availability Statement:** Data underlying the findings in this manuscript are available at https://doi.org/10.7910/DVN/TG31BU.

**Funding:** This project was funded by the Wellcome Trust (grants 220991/Z/20/Z and 203077/Z/16/Z), the Bill and Melinda Gates Foundation (INV-017547), and the Foreign Commonwealth and Development Office (FCDO) through the East Africa Research Fund (EARF/ITT/039) and is part of an integrated programme of SARS-CoV-2 sero-surveillance in Kenya led by KEMRI Wellcome Trust Research Programme. A.A. is funded by a DFID/MRC/NIHR/Wellcome Trust Joint Global Health Trials Award (MR/R006083/1), J.A.G.S. is funded by a Wellcome Trust Senior Research Fellowship (214320) and the NIHR Health Protection Research Unit in Immunisation, I.M.O.A. is funded by the United Kingdom's Medical Research Council and Department For International Development through an African Research Leader Fellowship (MR/S005293/1) and by the NIHR-MPRU at UCL (grant 2268427 LSHTM). G.M.W. is supported by a fellowship from the Oak Foundation. C.N.A. is funded by the DELTAS Africa Initiative [DEL-15-003], and the Foreign, Commonwealth and Development Office and Wellcome (220985/Z/20/Z). S.U. is funded by DELTAS Africa Initiative [DEL-15-003], L.I.O.-O. is funded by a Wellcome Trust Intermediate Fellowship (107568/Z/15/Z). R.A is funded by National Institute for Health Research (NIHR) (project reference 17/63/82) using UK aid from the UK Government to support global health research. The funders had no role in study design, data collection and analysis, decision to publish, or preparation of the manuscript. The views expressed in this publication are those of the authors and not necessarily those of the funding agencies.

**Competing interests:** The authors have declared that no competing interests exist.

## Results

We recruited 2,559 individuals from the three HDSS sites, median age (IQR) 27 (10–78) years and 52% were female. Seroprevalence at all three sites rose steadily during the study period. In Kisumu, Nairobi and Kilifi, seroprevalences (95% CI) at the beginning of the study were 36.0% (28.2–44.4%), 32.4% (23.1–42.4%), and 14.5% (9.1–21%), and respectively; at the end they were 42.0% (34.7–50.0%), 50.2% (39.7–61.1%), and 24.7% (17.5–32.6%), respectively. Seroprevalence was substantially lower among children (<16 years) than among adults at all three sites (p≤0.001).

## Conclusion

By May 2021 in three broadly representative populations of unvaccinated individuals in Kenya, seroprevalence of anti-SARS-CoV-2 IgG was 25–50%. There was wide variation in cumulative incidence by location and age.

## Introduction

Kenya has had five waves of SARS-CoV-2 infection as of January 2022. As of 31 December 2021 the official number of reported cases and deaths was 295,028 and 5,378 respectively [1]. The official number of reported cases is widely acknowledged to be an underestimate of the true number of infections. Accurate estimates of the proportion of the population previously infected with SARS-CoV-2 are important to understand the epidemiology of the infection and inform control strategies such as vaccination and non-pharmaceutical interventions. Previous studies have shown significant temporal and spatial heterogeneity in the pattern of infections in Kenya [2–6], but these studies had the limitation of not being population-based, instead relying on convenience samples of special population groups such as blood donors [2–6], ante-natal clinic (ANC) attendees,[5] truck drivers[4] and health care workers (HCW) [3]. A similar situation obtains in many low and middle income countries (LMICs), with only a handful of published population based studies having been conducted in Africa [7–11], which is home to over one billion people.

Sub-Saharan Africa appears to have had lower mortality from SARS-CoV-2 infections than other parts of the world. One reason that has been proposed for this is that the African population is generally younger [12]. To test this hypothesis, accurate age specific estimates of cumulative incidence of infection are required from population-based studies. These estimates can then be used to determine age-specific infection fatality rates. However, previous studies have been susceptible to several biases, including the absence of children in studies of blood donors, the restriction of ANC studies to females, and potentially higher exposure to SARS-CoV-2 among HCWs and truck drivers. In addition, many of these studies have over-sampled urban residents, yet majority of the people in sub-Saharan Africa live in rural areas.

We report results from the first round of a population-based survey for presence of SARS-CoV-2 antibodies at three [13–15] health and demographic surveillance sites broadly representative of the population of Kenya.

## Methods

### Study sites and participants

Following consultations with the Ministry of Health, three representative health and demographic surveillance systems (HDSS's) spread across Kenya were chosen as sites for the

serosurveillance study. The sites were as follows: Manyatta HDSS in Kisumu (Western Kenya) [13], Nairobi urban HDSS (Central Kenya) [14], and Kilifi HDSS [15], (South East Kenya) (S1 Fig). The total population under regular surveillance at the three sites is ~470,000 [13–15].

The Kisumu HDSS comprises a semi-urban population [13]. At the median date of sample collection in Kisumu (31 March 2021), the official number of reported cases of SARS-CoV-2 infection in the county was 2,670, and the official number of reported COVID-19 deaths was 24.

The Nairobi HDSS comprises an urban population [14]. At the median date of sample collection in Nairobi (28 March 2021), the official number of reported cases of SARS-CoV-2 infection in the county was 60,836, and the official number of reported COVID-19 deaths was 984.

The Kilifi HDSS comprises a predominantly rural population [15]. At the median date of sample collection in Kilifi (17 Feb 2021), the official number of reported cases of SARS-CoV-2 infection in the county was 2,613, and the official number of reported COVID-19 deaths was 21.

The predominant circulating SARS-COV-2 variants in Kenya at the median time when the serosurveys were done were the Alpha and Beta variants. (S2 Fig).

At each of the three sites, we aimed to recruit 850 randomly selected individuals across all age groups as follows: 100 individuals in each 5-year age band from 0–14 years, 50 individuals in each 5-year age band from 15–64 years and 50 individuals aged $\geq$ 65 years. A sample of 300 participants <15 years old would suffice to estimate seroprevalence of 1% with a 2% margin of error, and a sample of 550 participants aged $\geq$15 years would suffice to estimate seroprevalence of 3–5% with a <5% error margin.

Participants were selected from population lists of residents at the HDSS sites, which in the pre-pandemic period were updated at least twice a year. Surveillance activities at all sites were temporarily halted from March to September 2020 because of the disruption caused by the pandemic. Following approval from regulatory authorities and the Ministry of Health field-work activities resumed prior to the serosurvey, as this was deemed an essential public health activity.

Inclusion criteria for the study were that the individual should be a resident of the HDSS and provide consent. Potential participants were excluded if they had a bleeding disorder or other medical contraindication for venipuncture.

Members of the study team visited the homes of selected participants between 01 December 2020 and 27 May 2021. Potential participants that could not be found after three attempts were replaced by randomly selected individuals meeting the same age and sex category as the originally selected individual.

## Ethics and consents

Ethical approval for conducting the study and publication of these data was obtained from the Kenya Medical Research Institute Scientific and Ethics Review Unit (KEMRI/SERU/CGMR-C/203/4085). Written informed consent (and assent for individuals 13–17 years of age) was obtained from all participants, or their guardians (for individuals <18 years of age) before their participation in the study.

## Study procedures

Study staff administered a questionnaire to the participants either electronically (using android-run tablets) or on paper, to collect data on demographic and clinical characteristics, including whether the individual had experienced symptoms compatible with COVID-19 in

the preceding six months. None of the study participants had received any dose of a COVID-19 vaccine at the time of sample collection.

Study staff then collected 5ml (2ml in children under 5 years) of venous blood from each participant. Serum was obtained by centrifuging the samples at 450 x g for 5 minutes. Serum samples were then stored in -80˚C freezers at the study sites before being transported in dry ice to the KEMRI-Wellcome Trust research laboratories in Kilifi for assays.

**ELISA for SARS-CoV-2 spike protein.** All samples were tested at the KEMRI-Wellcome Trust Research Programme laboratories in Kilifi for IgG to SARS-CoV-2 whole spike protein using an adaptation of the Krammer Enzyme Linked Immunosorbent Assay (ELISA) [16]. Validation of the assay was described previously [6]. Briefly, sensitivity, estimated in 174 SARS-CoV-2 PCR positive recovering Kenyan COVID-19 patients in Nairobi and a panel of sera from the UK National Institute of Biological Standards and Control (NIBSC), was 92.7% (95% CI 87.9–96.1%); specificity, estimated in 910 serum samples from Kenya drawn in 2018 (i.e. pre-pandemic period), was 99.0% (95% CI 98.1–99.5). Results were expressed as the ratio of test OD to the OD of the plate negative control; samples with OD ratios greater than two were considered positive for SARS-CoV-2 IgG.

## Statistical analyses

In the primary analysis, we estimated seroprevalence at each site based on the weighted proportion of samples with an OD ratio >2. The study period was divided into six 30-day periods and separate seroprevalence estimates were computed for each period. For site specific age-stratified analyses, we used three time periods of 60 days each. We adjusted the seroprevalence estimates for assay sensitivity and specificity using both classical [17] and Bayesian modelling [18]. Summary estimates for each site were weighted to account for underlying population, age, and sex structure at the study sites. In the Bayesian modelling (code provided in S1 Text), non-informative priors were used for all parameters, and the models were fitted using the Rstan software package [19], as with previous studies [3,6]. We tested whether seropositivity was associated with sex, age, location, COVID-19 like symptoms, education level and period of sample collection using multivariable logistic regression.

All analyses were conducted using Stata™ Version 15 software (College Station, Texas, USA) and R version 3.6.1 (Vienna, Austria).

## Results

We recruited 2,559 individuals from the three health and demographic surveillance sites (Fig 1). The proportion of residents that provided consent to participate in the study in Kisumu, Nairobi, and Kilifi were 90%, 89% and 65% respectively. Samples were collected from 08 Feb 2021 to 27 May 2021 (median sample date 31 Mar 2021) in Kisumu, 21 Jan 2021 to 17 May 2021 (median sample date 28 Mar 2021) in Nairobi, and 01 Dec 2020 to 28 April 2021 (median sample date 17 Feb 2021) in Kilifi. The median age (IQR, range) of study participants was 27 years (10–78, 0–100) and 52% were female.

Fig 2 and S1 Table display Bayesian adjusted seroprevalence estimates at the three sites for each month between December 2020 and May 2021. In Kisumu seroprevalence was 36.0% (95 CI 28.2–44.4%) in February 2021 rising to 42.0% (95% CI 34.7–50.0%) in May 2021. In Nairobi, seroprevalence was 32.4% (95% CI 23.1–42.4%) in January 2021 rising to 50.2% (95% CI 39.7–61.1%) in May 2021. Seroprevalence in Kilifi rose substantially during the study period, starting at 14.5% (95% CI 9.1–21%) in December 2020, rising to 24.7% (95% CI 17.5–32.6%) in April 2021. The temporal differences in seroprevalence were statistically significant in the

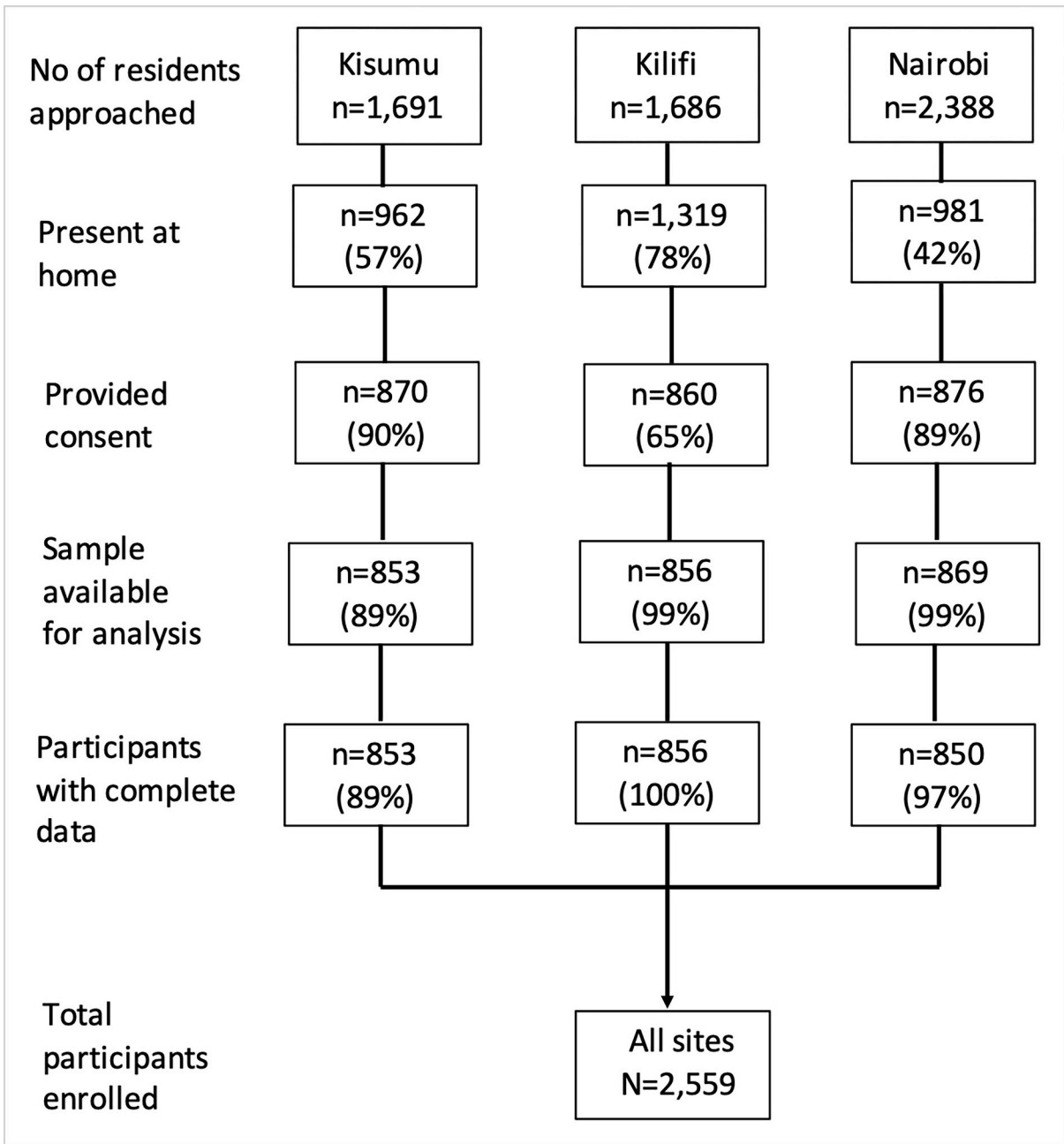

**Fig 1. Study flow chart.**

multivariable logistic regression models in Kilifi (p = 0.017) and Nairobi (p = 0.012) but not in Kisumu (p = 0.172).

Fig 3 displays Bayesian adjusted seroprevalence estimates for individuals ≥16 years and those ≤ 16 years for the entire study period at the three sites. Seroprevalence was substantially lower among children aged <16 years than among older individuals at each site. In Kisumu seroprevalence in individuals aged <16 years and ≥16 years was 30.1% (95% CI 24.4–36.3%) and 43.8% (95% CI 39.1–49%), respectively; in Nairobi, it was 29.1% (95% CI 23.7–35%) and 47.6% (95% CI 42.6–52.8%), respectively; in Kilifi it was 14.5% (95% CI 10.4–19.2%) and

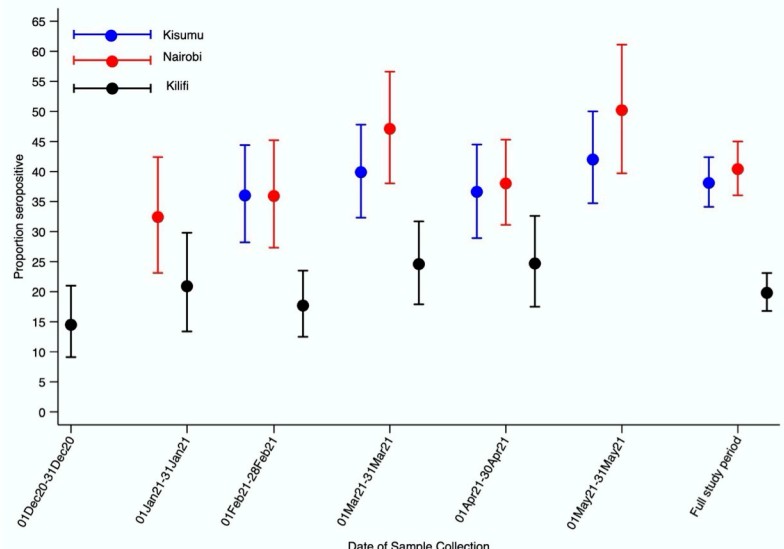

**Fig 2. Seroprevalence by HDSS site and study period.** Seroprevalence estimates were obtained using Bayesian threshold analysis adjusted for test performance and underlying population structure using multilevel regression and poststratification Error bars indicate 95% Credible Intervals.

25.1% (95% CI 21.1–29.3%), respectively. Variation in seroprevalence with age was significant at each of the three sites (P = 0.002 in Kisumu and P<0.001 Nairobi and Kilifi) and the pattern of variation was similar at each site. S2 Table displays Bayesian adjusted age specific seroprevalence in 10-year age bands at each of the three sites over three 60-day intervals.

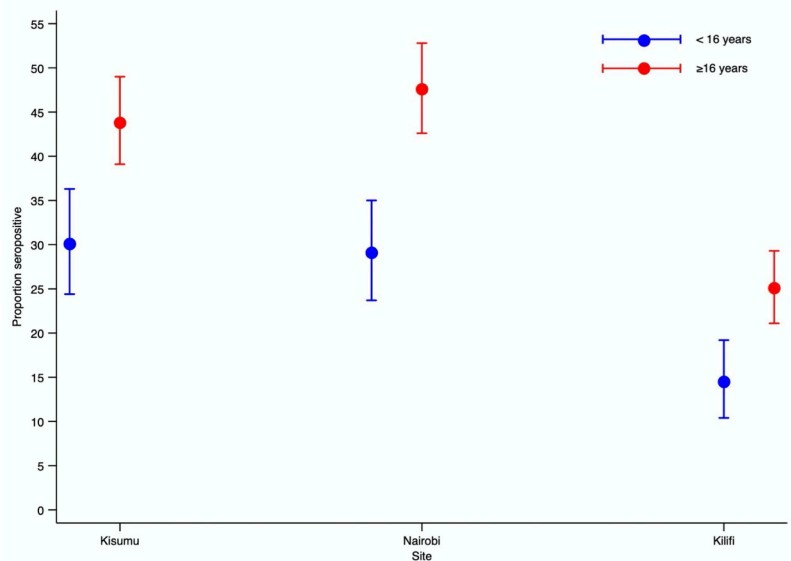

**Fig 3. Seroprevalence in children and adults by HDSS site.** Seroprevalence estimates were obtained using Bayesian threshold analysis adjusted for test performance and underlying population structure using multilevel regression and poststratification Error bars indicate 95% Credible Intervals.

Overall, seroprevalence in males (32.7%, 95% CI 29.6–36%) did not differ significantly from seroprevalence in females (33.8%, 95% CI 30.8–37.2%). Significant sex differences were not apparent in site specific analyses (S3 Table).

Results of analyses where we used classical methods to adjust for assay sensitivity and specificity were essentially similar to the Bayesian adjusted estimates (S3 and S4 Tables).

In a multivariate analysis testing for factors associated with seropositivity, we observed associations with age, study site and education level, but not with sex or history of symptoms (S5 Table).

## Discussion

In this population-based serosurveillance study using a locally-validated assay at three sites broadly representative of Kenya, we found that 25–50% of the resident populations had SARS-CoV-2 antibodies by the end of our study in May 2021. As none of the study participants had received a COVID-19 vaccine the presence of anti-spike antibodies indicates previous infection. There were large temporal, age-related and geographical differences in seroprevalence; in particular, older individuals and those in the urban areas of Kisumu and Nairobi had higher seroprevalence.

Our results are generally consistent with other community surveys conducted in Kenya at a roughly similar time point during the pandemic. In two household surveys conducted in Nairobi in November-December 2020, overall seroprevalence was 34.7% and 43.3% [10,20], where we found a seroprevalence of 32.4% in Nairobi in January 2021; we observed similar age related seroprevalence patterns compared to the previous surveys.

Our results from this population-based survey were also generally consistent with those obtained from special population groups in Kenya. This provides assurance that previous studies among other populations in Kenya have been broadly representative of seroprevalence trends. For example, seroprevalence among health care workers in Nairobi from July to December 2020 was 44% [3], compared to 32% in the general population of the Nairobi HDSS in January 2021. Seroprevalence among blood donors in the counties in which Kisumu, Nairobi, and Kilifi are located were 38%, 62% and 43% respectively in January to March 2021 [21], compared to 40%, 47% and 25% during March 2021 in the present study. Of note studies of blood transfusion donors are restricted to individuals aged 16–65 years who have substantially higher seroprevalence than children or older adults in our study. The lower seroprevalence observed in children is consistent with studies conducted in other parts of Africa [22] and may be due to age-related differences in the antibody response to infection with SARS-COV-2 [23]. Seroprevalence estimates from other population samples showed geographical variations of similar magnitude to those observed in the present study. Comparisons between different sampling methods are highly susceptible to confounding as illustrated above.

Our results once again demonstrate the significant discrepancy between officially reported numbers of cases at each of the study areas and actual population level exposure to SARS-COV-2 virus. The numbers of reported cases at each of the three study sites suggested that less than 1% of the population there had been exposed to the SARS-CoV-2 virus, in stark contrast to what we found on antibody testing. The disparate numbers are likely due to underreporting of cases as a results of limited public health infrastructure for morbidity and mortality surveillance, which has been highlighted previously [2,6,21]. The rapid rise in seroprevalence in Nairobi compared to Kilifi and Kisumu is consistent with studies of transmission dynamics that showed higher levels of population mixing in Nairobi compared to other less urbanized areas [24].

A significant strength of this study is the random sampling of a defined population. The results are therefore more generalizable to the Kenyan population than previous studies that involved special populations [2–6]. Although there had been two other population-based studies [10,20] conducted earlier, both were limited to Nairobi, the capital city of Kenya. Our inclusion of other sites in this study, far removed from Nairobi, helps to provide broader insights into the extent of the SARS-CoV-2 pandemic in Kenya. Another strength is that we used an assay that had been rigorously validated using samples from the local population and reference panels from the National Institute for Biological Standards and Control (NIBSC) in the UK, as well as the WHO [25,26].

A key limitation of the study is that we were unable to fully account for waning of antibodies, which could have led to an underestimation of infection prevalence. In a previous study, we undertook mixture modelling [27], which does not rely on thresholds, but instead assumes that the population sampled consists of two groups with different distributions of antibody levels; this suggested that the threshold-based method was significantly underestimating the true seroprevalence. We were however unable to conduct similar analysis for this dataset, as the mixture modelling requires control data from the respective sites which we did not have.

An additional limitation was the fact that nearly half of those who were targeted through the random sampling were not present at home. These were in most cases individuals who had gone to their workplaces. Given that these individuals would have had more interactions with others both at the workplace and during commuting, it is likely that the seroprevalence figures presented here are an underestimate of the true value.

In conclusion, in this population-based study at three health and demographic surveillance sites, broadly representative of the population in Kenya, we found that between a quarter and a half of the population had evidence of previous infection by May 2021, though with marked variation in infection risk by age and geographical region.

## Supporting information

**S1 Fig. Location of study sites within Kenya.** We used the Kenya boundaries outline shapefile from the Humanitarian data exchange platform https://data.humdata.org/dataset/cod-ab-ken. Population data were downloaded in raster format from WorldPop at 1km spatial resolution https://www.worldpop.org/.
(TIFF)

**S2 Fig. Timeline of COVID-19 waves in Kenya and serosurveys at the three health and demographic surveillance sites.** The orange data series are the daily number of new cases of COVID-19 in Kenya. The predominant variant behind each wave is denoted at the base of each wave. Data source: COVID-19 cases were obtained from Our World in Data (https://ourworldindata.org).
(TIFF)

**S3 Fig.**
(TIFF)

**S1 Table. Seroprevalence by HDSS site and study period.** [§] Figures represent percentages with 95% credible intervals in parentheses. Bayesian threshold analysis adjusted for test performance and underlying population structure using multilevel regression and poststratification
(DOCX)

**S2 Table. Age specific seroprevalence at three HDSS sites in Kenya.** ± Performance adjusted Bayesian threshold analysis adjusted for test performance and underlying population structure

using multilevel regression and poststratification. Figures represent percentages with 95% credible intervals in parentheses. § Kisumu data were collected from 01 Feb 2021–27 May 2021.
(DOCX)

**S3 Table. Sex-stratified seroprevalence by HDSS site and study period.** ± Kilifi HDSS population as at 18 Feb 2021 was used as the standard population. Prevalence estimates were adjusted for performance characteristics of assay used as described in methods section. § Figures represent percentages with 95% credible intervals in parentheses. Bayesian threshold analysis adjusted for test performance and underlying population structure using multilevel regression and poststratification.
(DOCX)

**S4 Table. Age and sex-stratified seroprevalence by HDSS site for entire study period.**
(DOCX)

**S5 Table. Multivariable analysis of factors associated with presence of antibodies to SARS-CoV-2 in Kisumu, Nairobi, and Kilifi.** § Per 5-year increase in age. ± Symptoms included the following: Abdominal pain, chest pain, cough, diarrhoea, fever and chills, headache, irritability and confusion, joint pains, loss of smell/taste, muscular pain, nausea and vomiting, runny nose, shortness of breath, sore throat and weakness.
(DOCX)

**S1 Text. Stan code for Bayesian adjustment of prevalence estimates to account for test performance and population structure.**
(DOCX)

## Acknowledgments

We thank Nelson Mbaya, Salma Musa, Boniface Ingumba Butichi and Nicholas Kiprono Mutai for their roles in data, laboratory and field work management in Nairobi. We thank Victor Akelo, Beth A. Tippet Barr, Dickson Gethi, Kephas Otieno and Frederick Oluoch for their technical support before and during implementation of the study in Kisumu. We thank F. Krammer for providing the plasmids used to generate the spike protein used in this work. Development of SARS-CoV-2 reagents was partially supported by the NIAID Centres of Excellence for Influenza Research and Surveillance (CEIRS) contract HHSN272201400008C. The COVID-19 convalescent plasma panel (NIBSC 20/118) and research reagent for SARS-CoV-2 Ab (NIBSC 20/130) were obtained from the NIBSC, UK. We also thank the WHO SOLIDARITY II network for sharing of protocols and for facilitating the development and distribution of control reagents. This paper has been published with the permission of the director, Kenya Medical Research Institute.

For the purpose of Open Access, the author has applied a CC-BY public copyright licence to any author accepted manuscript version arising from this submission.

## Author Contributions

**Conceptualization:** Anthony O. Etyang, Ifedayo Adetifa, James Nyagwange, Patrick Amoth, Kadondi Kasera, Amek Nyaguara, Sophie Uyoga, Katherine Gallagher, Eunice W. Kagucia, Dickens Onyango, Benjamin Tsofa, Joseph Mwangangi, Eric Maitha, Edwine Barasa, George M. Warimwe, J. Anthony G. Scott, Ambrose Agweyu.

**Formal analysis:** Boniface Karia, Mark Otiende, Christian Bottomley, Katherine Gallagher, Eunice W. Kagucia, Philip Bejon.

**Funding acquisition:** Anthony O. Etyang, Rashid Aman, Patrick Amoth, Mercy Mwangangi, Kadondi Kasera, Wangari Ng'ang'a, Lynette I. Ochola-Oyier, Sophie Uyoga, Eunice W. Kagucia, Edwine Barasa, Philip Bejon, George M. Warimwe, J. Anthony G. Scott, Ambrose Agweyu.

**Investigation:** Ifedayo Adetifa, Richard Omore, Thomas Misore, Abdhalah K. Ziraba, Maurine A. Ng'oda, Evelyn Gitau, Daisy Mugo, Monica Toroitich, James Tuju, Perpetual Wanjiku, Donald Akech, Antipa Sigilai, Angela Karani, Charles N. Agoti, Eric Maitha, Ambrose Agweyu.

**Methodology:** Anthony O. Etyang, Ifedayo Adetifa, Richard Omore, Thomas Misore, Abdhalah K. Ziraba, Maurine A. Ng'oda, Evelyn Gitau, John Gitonga, Daisy Mugo, Bernadette Kutima, Henry Karanja, Monica Toroitich, James Nyagwange, James Tuju, Perpetual Wanjiku, Donald Akech, Antipa Sigilai, Boniface Karia, Angela Karani, Charles N. Agoti, Lynette I. Ochola-Oyier, Amek Nyaguara, Sophie Uyoga, Eunice W. Kagucia, Dickens Onyango, George M. Warimwe, J. Anthony G. Scott, Ambrose Agweyu.

**Project administration:** Donald Akech, Shirine Voller, Amek Nyaguara, Eunice W. Kagucia, Dickens Onyango, Benjamin Tsofa, Joseph Mwangangi, Ambrose Agweyu.

**Supervision:** Maurine A. Ng'oda, Evelyn Gitau, Rashid Aman, Patrick Amoth, Mercy Mwangangi, Wangari Ng'ang'a, Angela Karani, Lynette I. Ochola-Oyier, Christian Bottomley, Amek Nyaguara, Dickens Onyango, Benjamin Tsofa, Joseph Mwangangi, Eric Maitha, Edwine Barasa, Philip Bejon, J. Anthony G. Scott, Ambrose Agweyu.

**Writing – original draft:** Anthony O. Etyang.

**Writing – review & editing:** Anthony O. Etyang, Ifedayo Adetifa, Thomas Misore, Abdhalah K. Ziraba, Maurine A. Ng'oda, Evelyn Gitau, John Gitonga, Bernadette Kutima, Henry Karanja, James Nyagwange, James Tuju, Mercy Mwangangi, Kadondi Kasera, Wangari Ng'ang'a, Donald Akech, Antipa Sigilai, Boniface Karia, Angela Karani, Shirine Voller, Charles N. Agoti, Lynette I. Ochola-Oyier, Mark Otiende, Christian Bottomley, Amek Nyaguara, Sophie Uyoga, Katherine Gallagher, Eunice W. Kagucia, Benjamin Tsofa, Eric Maitha, Edwine Barasa, Philip Bejon, George M. Warimwe, J. Anthony G. Scott, Ambrose Agweyu.

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
