## [Decision Letter · Decision Letter 0]

18 Apr 2022

PGPH-D-22-00247

SARS-CoV-2 Seroprevalence in three Kenyan Health and Demographic Surveillance Sites, December 2020-May 2021

Dear Dr. Etyang,

Thank you for submitting your manuscript to PLOS Global Public Health. After careful consideration, we feel that it has merit but does not fully meet PLOS Global Public Health’s publication criteria as it currently stands. Therefore, we invite you to submit a revised version of the manuscript that addresses the points raised during the review process.

Please submit your revised manuscript by . If you will need more time than this to complete your revisions, please reply to this message or contact the journal office at globalpubhealth@plos.org. Please include the following items when submitting your revised manuscript:

We look forward to receiving your revised manuscript.

Kind regards,

Hannah E. Clapham

Academic Editor

Journal Requirements:

1. Your co-authors:

Ifedayo Adetifa  -iadetifa@kemri-wellcome.org

Richard omore -omorerichard@gmail.com

Thomas Misore -tmisore@kemricdc.org

Abdhalah Ziraba -aziraba@aphrc.org

Evelyn Gitau -egitau@aphrc.org

John Gitonga -jgitonga@kemri-wellcome.org

Bernadette Kutima -BKutima@kemri-wellcome.org

Henry Karanja -HKaranja@kemri-wellcome.org

Monica Toroitich -mtoroitich@kemri-wellcome.org

James Tuju -JTuju@kemri-wellcome.org

Perpetual Wanjiku -PWanjiku@kemri-wellcome.org

Rashid Aman -amanra54@gmail.com

Patrick Amoth -patrickamoth@gmail.com

Mercy Mwangangi -mukuimwangangi@gmail.com

Wangari Ng'ang'a -nganga.wangari@gmail.com

Charles Agoti -CNyaigoti@kemri-wellcome.org

Lynette Ochola-Oyier -LiOchola@kemri-wellcome.org

Amek Nyaguara -ANyaguara@kemri-wellcome.org

Sophie Uyoga -SUyoga@kemri-wellcome.org

Katherine E Gallagher -KGallagher@kemri-wellcome.org

Eunice W Kagucia -ekagucia@kemri-wellcome.org

Benjamin Tsofa -BTsofa@kemri-wellcome.org

Eric Maitha -emaitha612@gmail.com

Edwine Barasa -EBarasa@kemri-wellcome.org

Philip Bejon -PBejon@kemri-wellcome.org

George M Warimwe -GWarimwe@kemri-wellcome.org

Ambrose Agweyu -AAgweyu@kemri-wellcome.org

,have not confirmed authorship of the manuscript. We have resent them the authorship confirmation email; however please check that the above email address for them is correct and follow up personally to ensure they confirm. 

Please note that we cannot proceed your manuscript  until we have received confirmations from all co-authors.

2. Please update the Funding Information tab in the system to matched with the Financial Disclosure.

3. Please update the completed 'Competing Interests' statement, including any COIs declared by your co-authors. If you have no competing interests to declare, please state "The authors have declared that no competing interests exist".

4. Please provide  separate figure files in .tif or .eps format only and remove any figures embedded in your manuscript file.  Please ensure that all files are under our size limit of 20MB.  

For more information about how to convert your figure files please see our guidelines: Once you've converted your files to .tif or .eps, please also make sure that your figures meet our format requirements

5. We notice that your supplementary [figures/tables] are included in the manuscript file.  Please remove them and upload them  with the file type 'Supporting Information'  . Please ensure that all Supporting Information files are included correctly and that each one has a legend listed in the manuscript after the references list. 

6. Please provide us with a direct link to the base layer of the map used in Fig S1 and ensure this location is also included in the figure legend. 

Please note that, because all PLOS articles are published under a CC BY license (creativecommons.org/licenses/by/4.0/), we cannot publish proprietary maps such as Google Maps, Mapquest or other copyrighted maps. If your map was obtained from a copyrighted source please amend the figure so that the base map used is from an openly available source.

Please note that only the following CC BY licences are compatible with PLOS licence: CC BY 4.0, CC BY 2.0  and CC BY 3.0, meanwhile such licences as CC BY-ND 3.0 and others are not compatible due to additional restrictions. If you are unsure whether you can use a map or not, please do reach out and we will be able to help you. 

The following websites are good examples of where you can source open access or public domain maps:

Additional Editor Comments (if provided):

Reviewers' comments:

Reviewer's Responses to Questions

**Comments to the Author**

1. Does this manuscript meet PLOS Global Public Health’s publication criteria? Is the manuscript technically sound, and do the data support the conclusions? The manuscript must describe methodologically and ethically rigorous research with conclusions that are appropriately drawn based on the data presented.

Reviewer #1: Yes

Reviewer #2: Yes

2. Has the statistical analysis been performed appropriately and rigorously?

Reviewer #1: Yes

Reviewer #2: Yes

3. Have the authors made all data underlying the findings in their manuscript fully available (please refer to the Data Availability Statement at the start of the manuscript PDF file)?

Reviewer #1: No

Reviewer #2: No

4. Is the manuscript presented in an intelligible fashion and written in standard English?

Reviewer #1: Yes

Reviewer #2: Yes

5. Review Comments to the Author

Reviewer #1: General comments:

This study provides useful data on SARS-CoV-2 seroprevalence from representative populations in Kenya, which adds to previous seroprevalence data from specific population subgroups (blood donors, truck drivers etc.) and includes data from both urban and rural settings. Available SARS-CoV-2 serological data are currently heavily skewed towards high-income settings and this study is a welcome addition to the literature. The analysis appears sound and the manuscript is for the most part clearly written. In some places the reader would benefit from greater detail, particularly in the methods, and the discussion section would be more compelling if the study findings were discussed in relation to the wider COVID-19 situation in the country.

Major comments:

1.Introduction – it would help for context if the authors could provide some information on the circulating SARS-CoV-2 variants during the time of the study

2.Methods – If district-level data on daily/weekly case counts are available, I would suggest including a figure that shows this for each study site, with an indication on when the data collection period occurs. This would give some better contextual information about when in the epidemic samples were collected and how much transmission was occurring. Currently, I don’t find the information on median sample dates and reported cases particularly informative.

3.Methods, p6, para 2 – I feel the first sentence could be re-written slightly to make the target number of participants clearer without the reader having to do unnecessary mental arithmetic. Personally, I don’t find the sample size calculation particularly necessary or useful – the hypothesised seroprevalence is much lower than reality and doesn’t take into account uncertainties introduced by post-stratification weighting or adjustments for assay performance. It’s clear from the results that the expected estimates and margins of error are quite irrelevant, and I think the uncertainty bounds from the Bayesian analysis speak for themselves.

4.Methods, p8, para 1 – please clarify when assent was applicable, as well as the age of consent before which parental consent was required.

5.Methods – the statistical analyses section needs some more detail to make it clear what specific analyses were done. At the very least, the models used to estimate seroprevalence correcting for assay sensitivity and specificity should be detailed in the appendix, together with the assumed priors, simulation conditions, and what method was used to determine the point estimate and confidence bounds. There’s also mention of a logistic model for factors associated with seropositivity, but it’s not clear if this was a Bayesian model or not. The findings from this model also don’t seem to be presented in the results section. Could you also clarify what is meant by “weighted to account for underlying population, age, and sex structure”? Were factors other than age and sex accounted for in the weighting? If so, what were they?

6.Results – could you please include the site-specific participation % in the first paragraph?

7.Table 1 – I feel this would be more compelling as a figure, perhaps showing the posterior estimates by time period as box/violin plots or something similar, and it would be easier to discern trends in relation to the uncertainty in the estimates. The table could be moved to the appendix.

8.Table 2 – in its present form, I don’t find this table to be very informative, because it’s clear that the survey wasn’t designed to provide reasonable estimates of seroprevalence within such fine age groupings. I also don’t think such fine age groupings are necessary, since epidemiologically, there’s not much reason to expect that 25-34 year-olds are very different to 35-44 year-olds in terms of SARS-CoV-2 infection risk. Children and the elderly are clearly different, but I would suggest combining adults into one or two categories. Presently, a number of these estimates are based on fewer than 10 individuals and the width of CrIs is larger than the point estimate in many cases. I would also be inclined to present this as a figure rather than a table (which again could be in the appendix), to allow for easier comparison across sites and age groups.

9.Results – in the methods, the authors say that information was collected on symptoms compatible with COVID-19. Was there any association between seropositivity and symptoms?

10.Discussion – it would be helpful to discuss the results in the context of the wider COVID-19 situation in the country, as well as comment on the possible reasons for differences in seroprevalence between sites. For example, how does the cumulative infection risk compare with the population risk of COVID-19 as determined by official statistics? And what are likely reasons why seroprevalence increased much more rapidly in Nairobi compared with the other two sites? It’s also interesting that seroprevalence in children was lower initially in Kilifi but rose dramatically over the study period. What might be the reasons for this? It’s not apparent in other age groups.

Minor comments:

1.P8, para 2 – please clarify what platform (software, hardware) was used for electronic data collection

2.P8, para 3 – “They” is ambiguous here, so suggest re-wording slightly.

3.P13, last para – antibody waning would lead to an underestimation of infection prevalence, rather than seroprevalence.

Reviewer #2: This is a well-conducted and nicely written SARS-CoV-2 seroprevalence survey from Kenya, which uses robust sampling methodology. It provides useful insight into the extent of community spread of SARSCoV2 up to end of May 2021 and is a helpful contribution to the literature. I have minor suggestions only.

1. Sampling: On page 7 it states potential participants that could not be found after 3 attempts were replaced by randomly selected individuals meeting same age and sex category as original individual. According to the Figure 1 flow chart (page 21) in Nairobi, of 2388 individuals approached, only 42% (981) were present at home. It would be helpful to know how many of the 2388 were from the original random selection. If less than 42% the sample becomes less random and more convenience based on who is at home. This has been addressed in the discussion, but it might be more transparent to add percentages in the 'Present at home' boxes of the flowchart.

2. Validation of the ELISA: the diagnostic test is well described and referenced but I found it difficult to establish what sort of patients sensitivity was evaluated on. Were the 174 SARSCoV2 PCR positive Kenyan adults all sick hospitalised patients? And any info re the panel from UK NIBSC? If they are broadly representative of recovered adults in the community then worth saying that. However if not, quoted assay sensitivity of 92.7% may be over-estimated (antibody titres will be higher if taken frm hospitalised patients soon after serious infection).

3. Context: It would be helpful to get a bit more context of the epidemic in Kenya. E.g. Was it predominatly alpha variant circulating between December 2020 and May 2021? In the intro it mentions there were 5 waves up to Jan 22 but when did deaths peak according to official figures? In the intro it states only 5378 deaths have been reported. Given such high pre-vaccine seroprevalence figures this is indicative of either a very low case fatality rate or massive under-reporting of COVID-related deaths. The fact that 55.5% of >65s were antibody positive in Nairobi but relatively few individuals died of COVID warrants some discussion.

4. The lower antibody prevalence in children also worth mentioning in the discussion. What do the authors think is reason for this? School closures? Or is it increased waning in this kids if they were the first to be affected in 2020? Is it possible that the lower blood volume in <5s (2ml vs 5ml) contributes to lower antibody positivity on the ELISA?

6. PLOS authors have the option to publish the peer review history of their article (what does this mean?). If published, this will include your full peer review and any attached files.

**Do you want your identity to be public for this peer review?** For information about this choice, including consent withdrawal, please see our Privacy Policy.

Reviewer #1: **Yes: **Clarence Tam

Reviewer #2: **Yes: **Barnaby Flower

---

## [Editor Report · Decision Letter 1]

15 Jul 2022

SARS-CoV-2 Seroprevalence in three Kenyan Health and Demographic Surveillance Sites, December 2020-May 2021

PGPH-D-22-00247R1

Dear Dr Etyang,

We are pleased to inform you that your manuscript 'SARS-CoV-2 Seroprevalence in three Kenyan Health and Demographic Surveillance Sites, December 2020-May 2021' has been provisionally accepted for publication in PLOS Global Public Health.

Best regards,

Hannah E. Clapham

Academic Editor